# Neutralizing and interfering human antibodies define the structural and mechanistic basis for antigenic diversion

Palak N. Patel[1], Thayne H. Dickey [1], Christine S. Hopp[2], Ababacar Diouf[3], Wai Kwan Tang[1], Carole A. Long[3], Kazutoyo Miura [3], Peter D. Crompton[2] & Niraj H. Tolia [1] ✉

Defining mechanisms of pathogen immune evasion and neutralization are critical to develop potent vaccines and therapies. Merozoite Surface Protein 1 (MSP-1) is a malaria vaccine antigen and antibodies to MSP-1 are associated with protection from disease. However, MSP-1-based vaccines performed poorly in clinical trials in part due to a limited understanding of the protective antibody response to MSP-1 and of immune evasion by antigenic diversion. Antigenic diversion was identified as a mechanism wherein parasite neutralization by a MSP-1-specific rodent antibody was disrupted by MSP-1-specific non-inhibitory blocking/interfering antibodies. Here, we investigated a panel of MSP-1-specific naturally acquired human monoclonal antibodies (hmAbs). Structures of multiple hmAbs with diverse neutralizing potential in complex with MSP-1 revealed the epitope of a potent strain-transcending hmAb. This neutralizing epitope overlaps with the epitopes of high-affinity non-neutralizing hmAbs. Strikingly, the non-neutralizing hmAbs outcompete the neutralizing hmAb enabling parasite survival. These findings demonstrate the structural and mechanistic basis for a generalizable pathogen immune evasion mechanism through neutralizing and interfering human antibodies elicited by antigenic diversion, and provides insights required to develop potent and durable malaria interventions.

Progress in reducing malaria morbidity and mortality has stalled[1], and emerging parasite resistance against existing drugs intensifies the need for alternative treatment strategies and preventive measures. A vaccine that targets malaria merozoites (or blood-stage parasites) would directly prevent parasite infection of red cells and clinical symptoms. To achieve a broadly protective blood-stage vaccine, it is crucial to identify essential and strain-transcending vaccine immunogens, the key epitopes that elicit potent neutralizing antibody responses, and the immune evasion mechanisms employed by the parasite to circumvent protection.

Merozoite surface proteins are high-priority candidate vaccine antigens as they are prime targets of the humoral immune response[2–4], of all such proteins, Merozoite Surface Protein 1 (MSP-1) is the most abundant, is essential for *Plasmodium* development[5,6] and is proposed to have a role in early erythrocyte attachment, invasion, and egress[7,8]. MSP-1 interactions with red cell proteins to facilitate these roles have

[1]Laboratory of Malaria Immunology and Vaccinology, National Institute of Allergy and Infectious Diseases, National Institutes of Health, Bethesda, MD, USA. [2]Malaria Infection Biology and Immunity Section, Laboratory of Immunogenetics, National Institute of Allergy and Infectious Diseases, National Institutes of Health, Rockville, MD, USA. [3]Laboratory of Malaria and Vector Research, National Institute of Allergy and Infectious Diseases, National Institutes of Health, Rockville, MD, USA. ✉e-mail: niraj.tolia@nih.gov

been well-characterized[9–11]. MSP-1 undergoes two distinct proteolytic processing steps (Fig. 1a) to first form 83, 30, 38, and 42 kDa fragments, followed by cleavage of the 42 kDa fragment into 33 and 19 kDa fragments[12]. The C-terminal p19 is attached to the merozoite surface through a GPI anchor and the remaining fragments are shed upon formation of a tight junction with the red blood cell (RBC). The structure of the ectodomain of MSP-1 lacking p19 revealed a concentration-dependent monomer–dimer equilibrium affected by the presence of red cell proteins which may compete for the dimerization interface[13]. p19 is maintained on the merozoite surface after invasion and thought to have a role in intraerythrocytic parasite development[14,15]. p19 consists of two epidermal growth factor (EGF)-like domains[3,16,17]. EGF-like domains are found in the extracellular domain of membrane-bound or secreted proteins[18] and serve a variety of functional roles including mediation of protein/protein interactions. The EGF-like domain includes six disulfide-bonded cysteine residues that stabilize a two-stranded beta-sheet connected to a second short, two-stranded sheet[18,19].

Antibodies targeting all MSP-1 subunits[20,21] can inhibit parasite growth to varying degrees and antibodies targeting p19 appear to be most potent[22]. Naturally acquired antibodies targeting p19 prevent merozoite invasion of RBCs and are associated with protection from clinical malaria[12,23–27] and protection appears to be FcγRI-mediated in a transgenic rodent malaria

model for MSP-1[28]. Monoclonal antibodies (mAbs) to MSP-1 isolated from rodents[17,22,29] and more recently from individuals with naturally acquired immunity[30,31] have been characterized. The murine mAb G17.12 binds the first EGF-like domain of p19 and does not inhibit erythrocyte invasion[17]. In contrast, murine mAbs 12.8 and 12.10 recognize overlapping epitopes on EGF domain 1 and inhibit erythrocyte invasion by preventing secondary proteolytic processing of MSP-1[22,29,32]. Recently, isolation of human IgG mAbs from individuals living in malaria-endemic areas with naturally acquired immunity identified three hmAbs, one of which, 42D6, showed potent activity in inhibiting parasite growth[30]. A separate study of human IgG identified the hmAb MaliM03 that binds a similar epitope as murine mAb G17.12 and likewise does not show any parasite growth inhibitory activity as an IgG[31]. Interestingly, forced multimerization of MaliM03 by incorporation into an IgM backbone achieved strong parasite binding and inhibited merozoite invasion of RBCs[31]. Immunization with p19 has been investigated in both animal[33,34] and clinical studies[35,36] as an approach induce protection. While p19-specific neutralizing antibodies were induced by a chimeric MSP vaccine in rabbits[34], phase 1 clinical trials of p19-based vaccines met with limited success[35,36]. In addition, various clinical studies have tested MSP-1-based vaccines, which were safe and elicited a humoral immune response[37–40]. However, limited effects on

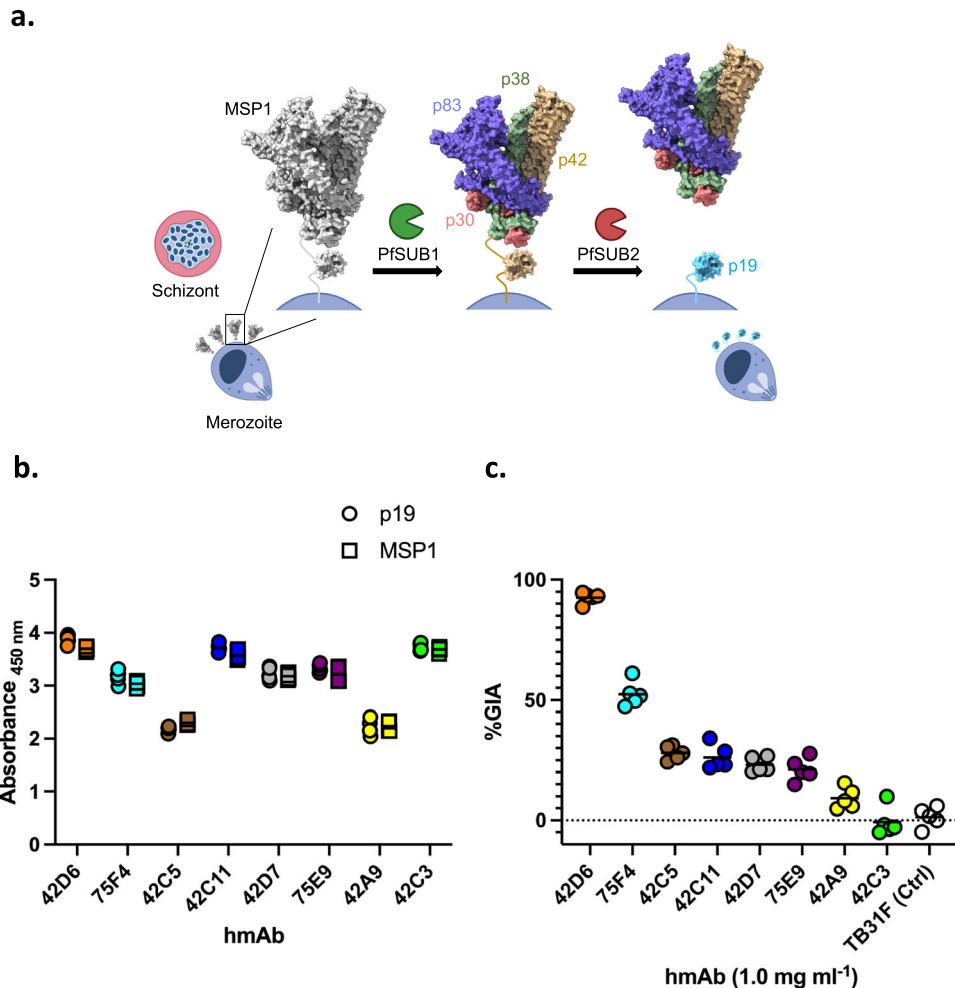

**Fig. 1 | Functional characterization of potent neutralizing Abs and high-affinity non-neutralizing Abs isolated from malaria-exposed individuals. a** Assembly and processing of the MSP-1. The figure was created using structure of MSP-1 (PDB ID: 6ZBF, https://www.rcsb.org/structure/6ZBF) in ChimeraX (https://www.rbvi.ucsf.edu/chimerax) and BioRender. **b** ELISA showing binding of the eight hmAbs to recombinantly expressed full-length MSP-1 and p19 from *Plasmodium falciparum 3D7* from 5 independent assays. **c** In vitro GIA of each hmAb tested at 1.0 mg/ml against the *Plasmodium falciparum 3D7* blood stage in five independent assays. The individual biological replicates from each assay and mean (bars) are shown. Source data are provided as a Source data file for Fig. 1b, c.

**Table 1 | Kinetic rate constants of binding for Fab fragments of eight human mAbs to p19, as determined by BLI**

| mAb | $K_D$ (×10$^{-9}$ ± SEM M) | $k_a$ (×10$^5$ ± SEM 1/Ms) | $k_{dis}$ (×10$^{-3}$ ± SEM 1/s) | N |
|---|---|---|---|---|
| 42D6 | 4.24 ± 0.01 | 2.12 ± 0.01 | 0.90 ± 0.01 | 3 |
| 75F4 | 207.87 ± 6.97 | 3.00 ± 0.07 | 62.21 ± 0.61 | 3 |
| 42C5 | 3.29 ± 0.05 | 2.22 ± 0.02 | 0.73 ± 0.01 | 3 |
| 42C11 | 1.84 ± 0.03 | 2.55 ± 0.01 | 0.47 ± 0.01 | 3 |
| 42D7 | 6.20 ± 0.08 | 2.80 ± 0.01 | 1.73 ± 0.02 | 3 |
| 75E9 | 300.17 ± 14.52 | 1.46 ± 0.04 | 43.79 ± 0.99 | 3 |
| 42A9 | 0.73 ± 0.01 | 2.76 ± 0.01 | 0.20 ± 0.01 | 3 |
| 42C3 | 0.66 ± 0.01 | 2.31 ± 0.01 | 0.15 ± 0.01 | 3 |

Dissociation constant ($K_D$), association rate constant ($k_a$), and dissociation rate constant ($k_{dis}$).

parasite growth rates in the blood and limited efficacy were observed[38,39].

Antigenic diversion has been observed for antibodies that target MSP-1, whereby non-inhibitory anti-parasite antibodies prevent the activity of a potent rodent antibody[24,32,41,42]. Such non-inhibitory antibodies have been called "blocking" antibodies in the past[32,41], however, we recommend the term "interfering" antibodies to avoid confusion with receptor-blocking antibodies. Blocking/interfering antibodies have been proposed to function by preventing inhibitory mAbs from binding or inhibiting MSP-1 through diverse mechanisms[32,41]. Antigenic diversion has been observed with polyclonal rabbit anti-MSP-1 antibodies that bind epitopes outside of p19, anti-MSP-1 mouse mAbs that bind epitopes within p19, and for affinity-purified, naturally acquired human antibodies specific for epitopes within the 83-kDa domain of MSP-1[32,41].

Here, we characterize a panel of naturally acquired p19-specific hmAbs through a combination of structural studies, parasite growth disruption, and biophysical analysis. We determine the co-crystal structures of p19 bound to a potent, broadly neutralizing hmAb and to non-neutralizing hmAbs (blocking/interfering antibodies). This study provides insights into the mechanism of antigenic diversion whereby blocking/interfering antibodies occlude the epitope targeted by neutralizing antibodies. Finally, we elucidate how p19-specific neutralizing antibodies can protect an individual from malaria parasite infection and identify key epitopes to guide future structure-based vaccine design.

## Results

### Production of p19-specific human monoclonal antibodies, p19, and full-length MSP-1

A panel of MSP-1-specific IgG$^+$ B cell receptor sequences was generated from adult volunteers enrolled in an observational cohort study conducted in the malaria-endemic community of Kalifabougou, Mali[30]. These sequences were cloned into human immunoglobulin G1 (IgG1), kappa (k), or lambda (λ) scaffolds to produce recombinant hmAbs. Paired heavy- and light-chain plasmids were co-expressed (Supplementary Fig. 1a), and antigen specificity was confirmed by ELISA reactivity of purified recombinant hmAbs to p19 and full-length MSP-1 (Fig. 1b). The expressed non-glycosylated p19 and full-length MSP-1 were monomeric and monodisperse, as observed by size-exclusion chromatography and SDS-PAGE (Supplementary Fig. 1b and c) Eight of the MSP-1-specific hmAbs generated bound to p19. Six of these eight antibodies were isolated from one individual (42) and the other two from a second individual (75) (Supplementary Table 1).

### Binding kinetics characterization of isolated antibodies

Antibody affinity and binding kinetics may be important determinants for protection and neutralization. We determined the binding kinetics of eight antigen-binding fragments (Fabs) to p19 by Biolayer Interferometry (BLI). We observed a range of dissociation constants ($K_D$)

from 0.66 to 300 nM (Table 1 and Supplementary Fig. 2). hmAbs 42C3 and 42A9 bound to p19 with the strongest affinities and 42C11 bound ~2–3-fold weaker. hmAbs 42C5 and 42D6 bound with moderate affinity ~6-fold weaker than 42C3, and 42D7 bound ~10-fold weaker than 42C3. Finally, 75E9 and 75F4 bound more than 300-fold weaker than 42C3. These diverse binding affinities for p19 derive predominantly from varied dissociation rates ranging from $0.15 \times 10^{-3}$ to $62.21 \times 10^{-3}$ s$^{-1}$ while association rates are relatively consistent between antibodies (Table 1).

### 42D6 potently neutralizes blood-stage parasites

We evaluated the ability of hmAbs to block the entry of merozoites into erythrocytes using the standardized growth inhibition activity (GIA) assay (Fig. 1c). hmAb 42D6 inhibited parasite growth by >90% (against *Plasmodium falciparum 3D7*) at 1.0 mg/ml, and had a binding affinity of 4.24 nM. 42C3 and 42A9 had stronger binding affinities than 42D6 but showed no inhibition of parasite growth at 1.0 mg/ml in GIA. These data establish that there is no correlation between binding kinetics and GIA (Supplementary Fig. 3), and that binding kinetics alone are insufficient to predict inhibition of erythrocyte invasion by merozoites.

### Epitopes for p19-specific hmAbs are overlapping

Epitope binning revealed that the panel of eight p19 hmAbs all compete with one another for binding by biolayer interferometry (BLI). Six of the eight Fabs had slow dissociation rates suitable for use as the primary antibody, and all eight Fabs were suitable for use as the secondary or competing antibody. Strikingly, all hmAbs competed with one another suggesting their epitopes are either adjacent or overlapping (Fig. 2a).

### Structure of p19 in complex with Fab fragments 42D6, 42C11, and 42C3

The lack of correlation between GIA and antibody affinity or epitope-binning prompted a comprehensive structural analysis to map the epitopes of inhibitory and non-inhibitory antibodies. We determined co-complex crystal structures of p19 with 42D6, 42C11, and 42C3 to resolutions of 2.0, 1.9, and 2.3 A° (Fig. 2b and Supplementary Fig. 4, Table 2). 42C11 and 42C3 were selected for further study due to their diverse growth inhibitory potential and diverse binding affinity.

The structures revealed that 42D6 binds p19 via heavy-chain interactions with residues at the central β-sheets and C-terminal loop of EGF-like domain II and a few interactions with residues from the N-terminal loop and the loop C-terminal to the central β-sheets of EGF-like domain I (Supplementary Fig. 5a). 42D6 has an interacting buried surface area (BSA) of 710.0 Å$^2$ with p19, and residues from all three CDR loops of the heavy chain contact twenty-one p19 residues (Supplementary Table 2). The conformational epitope recognized by the hmAb 42D6 on p19 does not overlap with epitopes for non-neutralizing hmAb MaliMO3 (PDB ID: 6XQW, https://www.rcsb.org/structure/6XQW)[31] or murine mAb G17.12 (PDB ID: 1OB1, https://www.rcsb.org/structure/1OB1)[17] (Fig. 2c).

In contrast to the 42D6 epitope, co-crystal structures of non-neutralizing hmAbs 42C11 and 42C3 revealed that both hmAbs primarily recognize EGF-like domain I while making few contacts with EGF-like domain II (Supplementary Fig. 5b and c). The epitopes for 42C11 and 42C3 are largely overlapping (Fig. 2b, c) with large interacting BSAs of 876.8 and 902.7 Å$^2$, respectively. In addition, both their heavy- and light-chains contributed almost equally to p19 contacts and BSA (Supplementary Tables 3 and 4, respectively).

### 42C11 and 42C3 represent an immunodominant antibody lineage

The shared epitope of 42C11 and 42C3 is consistent with their high sequence similarity, including similar CDR3 sequences and shared

**a.**

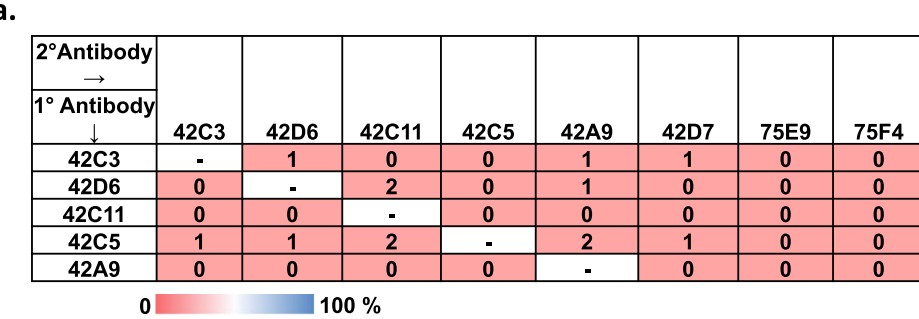

| 2°Antibody → 1° Antibody ↓ | 42C3 | 42D6 | 42C11 | 42C5 | 42A9 | 42D7 | 75E9 | 75F4 |
|---|---|---|---|---|---|---|---|---|
| 42C3 | - | 1 | 0 | 0 | 1 | 1 | 0 | 0 |
| 42D6 | 0 | - | 2 | 0 | 1 | 0 | 0 | 0 |
| 42C11 | 0 | 0 | - | 0 | 0 | 0 | 0 | 0 |
| 42C5 | 1 | 1 | 2 | - | 2 | 1 | 0 | 0 |
| 42A9 | 0 | 0 | 0 | 0 | - | 0 | 0 | 0 |

0 ▬▬▬ 100 %

**b.**

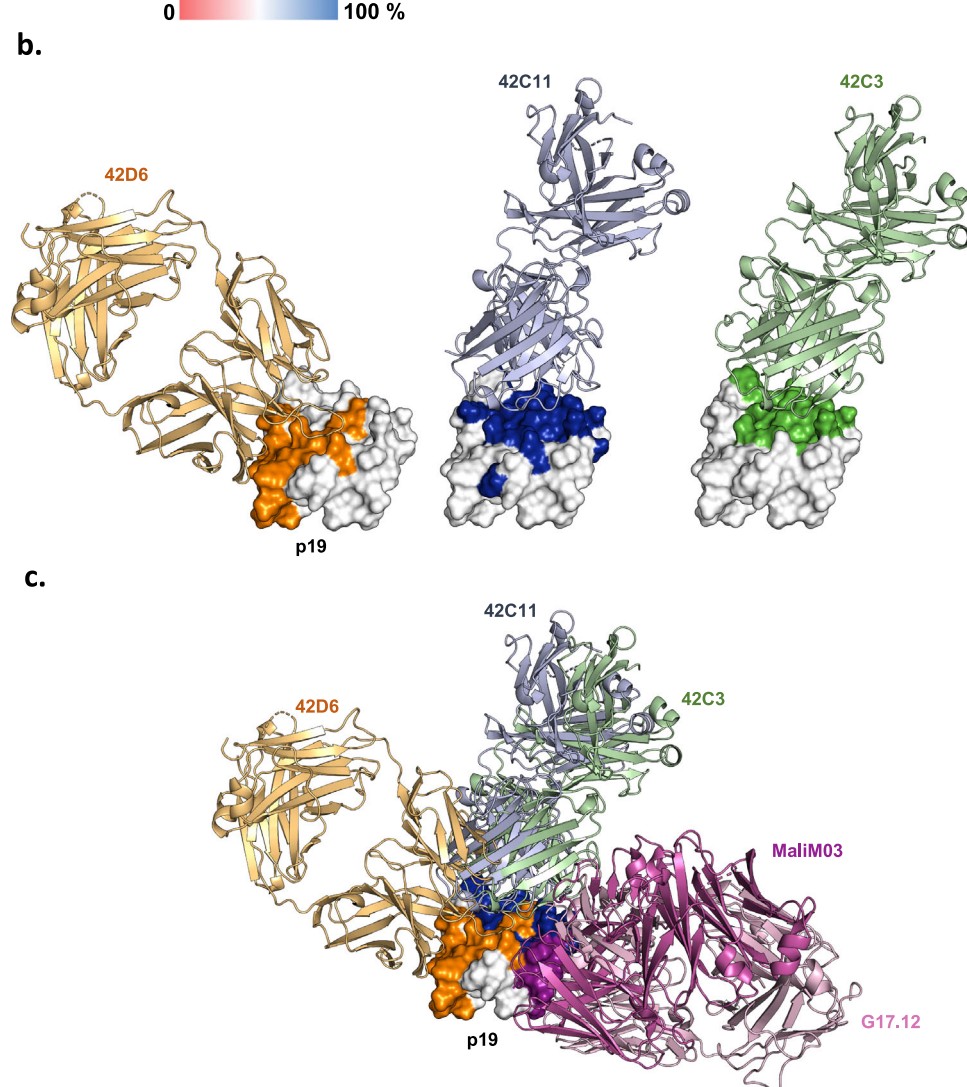

**c.**

**Fig. 2 | 42D6 is a potent strain-transcending neutralizing Ab targeting a distinct epitope that partially overlaps with sites bound by non-neutralizing Abs.** **a** Epitope binning for the p19-specific human mAbs. Primary saturating antibodies tested are listed in the left column, while secondary competing antibodies are listed at the top in rows. Data indicate the percent of competing antibody binding compared to the maximum competing antibody response in the absence of the primary antibody. Boxes are colored according to competition status. Antibodies that displayed ≤50% maximal binding are colored light red and are considered "competing". Negative values were normalized to 0. **b** Crystal structures of p19 bound to the Fab fragment of 42D6 (bright orange), 42C11 (light blue), and 42C3 (light green). All crystal structures are shown according to the same p19 orientation. p19 is represented as white surface. **c** Structural basis for the inhibition of parasite growth by p19-specific hmAb 42D6. Superposition of p19-Fab co-complex crystal structures. p19 is represented as white surface and Fabs are shown as cartoon representation and colored as in **b**. The epitope for non-neutralizing hmAb MaliM03 (PDB ID: 6XQW) and murine mAb G17.12 (PDB ID: 1OB1) is colored in magenta and Fab fragment of MaliM03 and G17.12 are colored in light magenta and light pink, respectively. The potent neutralizing hmAb 42D6 is shown to recognize a novel epitope on p19.

heavy- and light-chain germlines. These antibodies, in addition to the similar 42C5, 42D7, and 42A9, were isolated from the same individual and are likely clonally related (Supplementary Fig. 6). Interestingly, 6/29 of the MSP-1-specific heavy-chain IgG sequences isolated from this individual had highly similar sequences suggesting significant clonal expansion of this antibody lineage. All these hmAbs were poorly neutralizing and competed with all other hmAbs (Figs. 1c and 2a). This lineage utilizes the IGHV3-30 germline, which is the most common

**Table 2 | Crystallography data and refinement statistics**

| | p19-42D6 Fab complex (PDB ID: 8DFG) | p19-42C11 Fab complex (PDB ID: 8DFI) | p19-42C3 Fab complex (PDB ID: 8DFH) |
|---|---|---|---|
| **Data collection** | | | |
| Space group | P 1 2₁ 1 | P 1 2₁ 1 | P2₁2₁2₁ |
| **Cell dimensions** | | | |
| a, b, c (Å) | 41.68, 71.37, 196.50 | 62.56, 68.77, 63.73 | 72.39, 75.72, 117.53 |
| a, b, g (°) | 90.00, 94.64, 90.00 | 90.00, 110.68, 90.00 | 90.00, 90.00, 90.00 |
| Resolution (Å) | 19.7–1.998 (2.07–1.998) | 19.36–1.898 (1.966–1.898) | 19.83–2.297 (2.379–2.297) |
| R-meas (%) | 12.7 (80.9) | 5.2 (37.0) | 8.9 (74.4) |
| I/σ | 6.03 (1.61) | 18.85 (4.86) | 10.99 (1.98) |
| CC₁/₂ | 99.4 (76.6) | 99.9 (90.3) | 99.6 (78.3) |
| Completeness (%) | 96.26 (92.26) | 94.78 (78.36) | 96.72 (94.80) |
| Redundancy | 3.25 | 3.45 | 3.68 |
| No. of complex/ASU | 2 | 1 | 1 |
| **Refinement** | | | |
| Resolution (Å) | 19.915–1.998 | 19.849–1.898 | 19.829–2.297 |
| No. of reflections | 75098 | 37949 | 28461 |
| R-work/R-free | 0.2249/0.2660 | 0.1752/0.2001 | 0.2093/0.2329 |
| **No. of atoms and residues** | | | |
| Protein residues | 1040 | 515 | 521 |
| Wilson B-factor | 29.60 | 27.26 | 47.38 |
| Protein | 7985 | 3903 | 3943 |
| Water | 341 | 269 | 51 |
| **r.m.s. deviation** | | | |
| Bond lengths (Å) | 0.010 | 0.004 | 0.002 |
| Bond angles (Å) | 0.93 | 0.70 | 0.57 |
| **Validation** | | | |
| MolProbity score | 0.97 | 0.81 | 0.79 |
| Clashscore | 2.43 | 1.17 | 1.16 |
| Poor rotamers (%) | 0.00 | 0.00 | 0.00 |
| **Ramachandran plot** | | | |
| Favored (%) | 97.75 | 97.02 | 97.47 |
| Allowed (%) | 2.25 | 2.98 | 2.53 |
| Outliers (%) | 0.00 | 0.00 | 0.00 |

germline across multiple individuals for MSP-1/AMA1-specific B cells[30]. Together, these observations suggest that 42C11 and 42C3 represent a clonally expanded immunodominant antibody lineage.

### 42D6 targets a conserved epitope on p19

Antigen polymorphism can potentially limit strain-transcending protection by vaccine-induced or naturally acquired antibodies and should be evaluated in the context of hmAb binding and neutralization. We structurally mapped polymorphic residues within p19 identified from 3488 amino acid sequences in the MalariaGEN Pf3k database (Fig. 3a) (https://www.malariagen.net). Nineteen 42D6 epitope residues in p19 were invariant and two residues exhibited polymorphisms with varying frequencies [Glu65Lys (0.5%), and Leu86Phe (19.0%)]. Thr61, which is in the vicinity of the 42D6 epitope, also exhibited polymorphism (Thr61Lys) with an observed frequency of 76.6%. None of the polymorphisms had a major effect on binding with Thr61Lys and Leu86Phe decreasing affinity less than 3-fold, and the rare Glu65Lys polymorphism decreasing affinity 6-fold (Fig. 3b and Supplementary Table 5). These data suggest the 42D6 epitope is broadly conserved and 42D6 recognizes a wide array of p19 variants.

### 42D6 is a strain-transcending broadly-neutralizing human mAb

We further examined the strain-transcending neutralizing potential of 42D6 by performing GIA assays against three diverse strains of *Plasmodium falciparum*: 3D7, Dd2, and FVO. These strains contain high-frequency polymorphisms within p19 and form a strong foundation to evaluate the breadth of 42D6. p19 from Dd2 possesses three polymorphisms relative to 3D7: Thr61Lys, Ser70Asn, and Arg71Gly; and p19 from FVO possesses four polymorphisms: Gly1Gln, Thr61Lys, Ser70Asn, and Arg71Gly (Supplementary Fig. 7). All strains were neutralized by 42D6 with half-maximum inhibitory concentration ($IC_{50}$) values of 0.106, 0.259, and 0.317 mg/ml against *P. falciparum* 3D7, FVO, and Dd2 strains, respectively (Fig. 3c). These data indicate that 42D6 is a likely to be a broadly neutralizing human antibody.

### High-affinity non-neutralizing antibodies against the adjacent or overlapping region interfere with the effect of neutralizing antibodies (antigenic diversion)

We have established that 42D6 is a potently neutralizing p19-specific hmAb that targets a unique epitope. In addition, we also identified epitopes for two non-neutralizing hmAbs 42C3 and 42C11, which partially overlap with the neutralizing epitope of 42D6 hmAb. These diverse hmAb parameters and functions prompted the question of how these hmAbs may interact or interfere with each other and modulate parasite survival. We examined the interactions of these hmAbs using combination GIA assay to evaluate potential effects.

Strikingly, combining 42C3 with 42D6 completely abrogated the ability of 42D6 to neutralize parasites (Fig. 4a). Similarly, combining 42C11 with 42D6 reduced GIA inhibition to a level similar to 42C11 alone (Fig. 4a). These data are consistent with the non-neutralizing high-affinity antibodies 42C11 and 42C3 preventing binding of 42D6, thereby enabling parasite survival (Fig. 4b, c).

### Discussion

Naturally acquired antibodies that bind to p19 are found in individuals from malaria-endemic regions and have been associated with reduced morbidity[26,27]. While hmAbs to p19 have been recently identified, information on their breadth and potency and their impact on immune evasion mechanisms including antigenic diversion are very limited. Here, we structurally and functionally characterize the naturally acquired human antibody response to p19.

Isolated hmAbs were analyzed in a series of integrated approaches to evaluate binding affinity, neutralization potential, and structure-function relationships. We demonstrate the hmAb 42D6 is a potent strain-transcending neutralizing hmAb with an IC50 of approximately 0.106 mg/ml in GIA assays. Polymorphic variant analysis revealed that the 42D6 epitope is largely conserved, that 42D6 can bind all sequence polymorphisms with nanomolar affinity, and that 42D6 can inhibit parasite growth of diverse strains. 42D6 could be assessed as a malaria prophylactic either alone or in combination with other hmAbs.

Combinatorial GIA assays and p19 co-crystal structures clearly show that anti-p19 hmAbs that do not inhibit erythrocyte invasion can interfere with the inhibitory activity of potent neutralizing hmAbs (Fig. 4a, b). Binding kinetics and epitope binning data (Table 1 and Fig. 2a) suggest that these naturally acquired non-neutralizing hmAbs function by competing with neutralizing hmAbs for a single site on the merozoite. The structural, functional, and mechanistic data provide direct evidence of antigenic diversion, with the proof-of-concept that high-affinity interfering hmAbs 42C3 and 42C11 abolish or reduce the biological activity of potent neutralizing hmAb 42D6. Antigenic diversion has been previously observed with polyclonal rabbit anti-MSP-1 antibodies that bind epitopes outside of p19, and for affinity-purified, naturally acquired human antibodies specific for epitopes within the 83-kDa domain of MSP-1[41].

When antigenic diversion is prevalent, the protective potential of a p19-based vaccine could be impaired by pre-existing or vaccine-

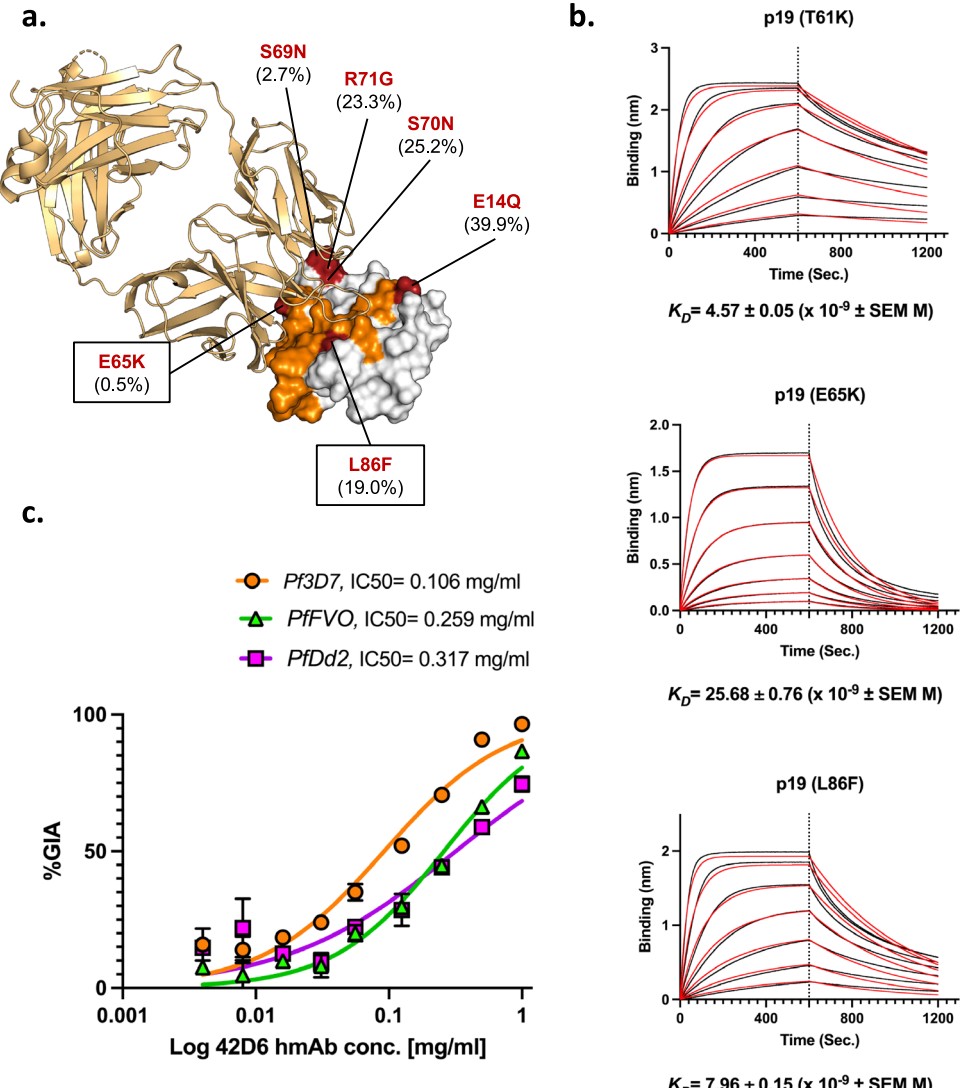

**Fig. 3 | Human mAb 42D6 targets a conserved epitope on p19. a** p19 sequence polymorphisms. Polymorphism mapped onto the p19 surface. 42D6 epitope is colored in orange. Polymorphic residues are shown in red and the observed substitutions as well as the percent minor allele frequency (MAF) of substitutions are indicated. p19 is represented as white surface and 42D6 Fab fragment as bright orange cartoon. **b** Binding affinity of 42D6 Fab to p19 constructs with point mutations representative of sequence polymorphisms in the 42D6 epitope as measured by BLI and source data are provided as a Source data file. **c** Cross-neutralization potential of 42D6. In vitro GIA dilution series against the *Pf3D7* reference strain, *PfFVO*, and *PfDd2* strains. $IC_{50}$ values were determined by interpolation after fitting data to a four-parameter dose-response curve. Data are plotted as the mean ± standard deviation and source data are provided as a Source data file.

induced interfering antibody responses to p19 or the MSP-1 complex. The expansion of the interfering B cell lineage described here suggests that interfering antibodies may be abundant, contributing to their ability to outcompete neutralizing hmAbs. Furthermore, the prevalence of the IGHV3-30 germline in MSP1-specific B cells identified in other individuals suggests this response may be a common, or "public", antibody response[30]. Immunizing an individual with an abundant pre-existing reservoir of interfering memory B cells would be expected to stimulate an interfering antibody response, rather than the desired neutralizing response. One potential approach to overcome antigenic diversion is the structure-guided design of immunogens that selectively elicit neutralizing antibodies in malaria-exposed individuals and/or effectively elicit neutralizing, but not interfering, antibodies in naïve individuals.

The findings delineate novel epitopes targeted by naturally acquired antibodies and provide proof-of-concept of antigenic diversion, as evidenced by epitope binning, combinatorial GIA, and co-

crystal structures of p19 that clearly showed that non-neutralizing antibodies competitively prevent binding of neutralizing antibodies to p19 on the merozoite surface. These findings provide a structural and mechanistic basis for previous reports which showed that interfering or counter-neutralizing antibodies can be induced by natural exposure to malaria infection[41]. Antigenic diversion may be a generalizable phenomenon applicable to other pathogens that can evade the immune response by simply exploiting the binding attributes of human antibodies. This study may help inform suboptimal antibody protection observed in the context of other infectious diseases and vaccines.

## Methods
### Samples and ethical approval
The hmAbs characterized in this study were isolated from PBMCs obtained from subjects enrolled in an observational cohort study conducted in the rural community of Kalifabougou, Mali. Details of the

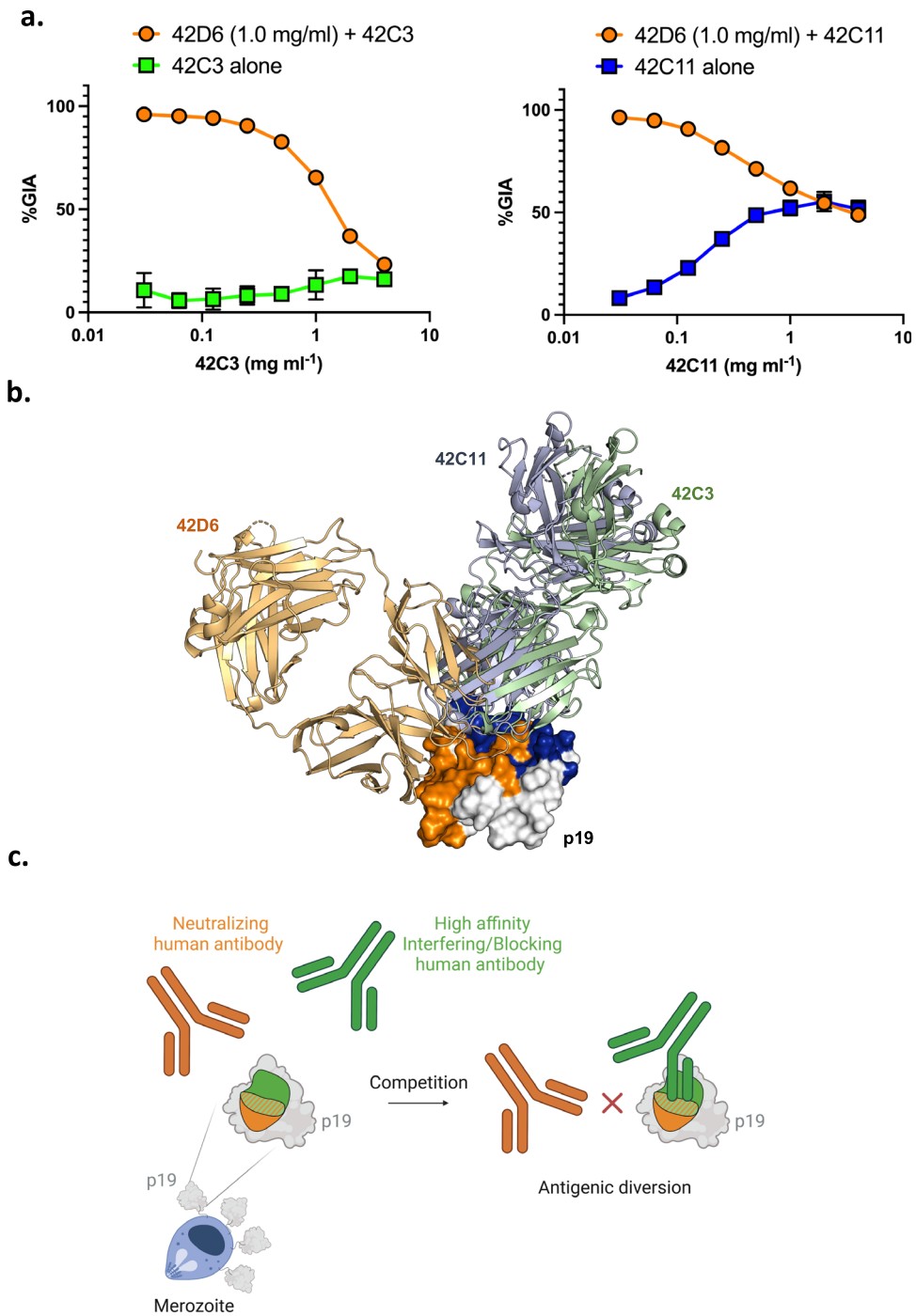

**Fig. 4 | Structural basis of antigenic diversion. a** GIA of neutralizing hmAb 42D6 (1.0 mg/ml) in the presence of an increasing concentration of non-neutralizing hmAbs 42C3 and 42C11 that block its binding to p19. The data arise from three independent biological replicates and plotted as mean ± standard deviation. Source data are provided as a Source data file. **b** Superposition of p19-Fab co-complex crystal structures. p19 is represented as white surface and Fabs are shown as cartoon representation and colored as in Fig. 2b. **c** High affinity interfering/blocking hmAb block the binding of potent neutralizing hmAb proving the concept of antigenic diversion. The figure was created in BioRender.

study cohort, sample processing, and hmAb isolation have been described previously[30,43]. The Ethics Committee of the Faculty of Medicine, Pharmacy, and Dentistry at the University of Sciences, Technique, and Technology of Bamako, and the Institutional Review Board of the National Institute of Allergy and Infectious Diseases, National Institutes of Health, approved this study. Written informed consent was obtained from adult participants and from the parents or guardians of participating children. The cohort study is registered in the ClinicalTrials.gov database (NCT01322581).

## Single-cell BCR sequencing and cloning of hmAbs

BCR sequencing was previously reported[30]. Briefly, RT-PCR was used to amplify the heavy- and light-chain variable regions of single IgG+ antigen-specific B cells from five de-identified donors. PCR products were Sanger sequenced and analyzed using IMGT/V-QUEST to identify complete heavy- and light-chain pairs with high-quality reads covering all CDRs[44]. Thirty-four paired sequences had sequences suitable for recombinant expression. Codon-optimized variable region sequences were fused to the human IGHG*01, IGKC*01, or IGLC2*02 constant

regions and cloned into the pHL-sec plasmid (GenScript)[45]. mAbs were produced via transient transfection of HEK293 cells and screened for p19 binding via ELISA. Multiple sequence alignments were constructed using Clustal Omega/T-Coffee[46].

### Expression, purification, and characterization of p19 and full-length MSP-1

The 3D7 allele of p19 and full-length MSP-1 were expressed in HEK293 cells, a system capable of post-translational modifications. Recombinant protein production in mammalian cells runs the risk of Asn x Thr/Ser N-linked sites being glycosylated when such sites are not glycosylated in *Plasmodium*. There are two putative N-linked sites in the primary amino acid sequence of p19 and fifteen putative N-linked sites in the full-length MSP-1. p19 and full-length MSP-1 sequences were codon optimized for expression in mammalian cells (GenScript) and all N-linked glycosylation sites (NXS/T) were modified by substituting the serine or threonine residue with an alanine residue to prevent the glycosylation that is absent in the endogenous *P. falciparum* proteins.

These optimized coding sequences were cloned into a pHL-sec vector which incorporates $His_6$ tag to the C-terminal[45] and transfected into Expi293F™ cells (Thermo Fisher Scientific, Cat# A14527). The soluble proteins were purified from cell-free supernatant 4–5 days post-transfection using Ni Sepharose® Excel resin (Cytiva) and size exclusion chromatography (Superdex 75 Increase 10/300 GL; Cytiva) in a phosphate buffered saline or 20 mM Tris (pH 8.0) containing 100 mM NaCl. Size exclusion chromatography was performed on a ÄKTA pure protein purification system and data was collected using UNICORN 7.3 software.

To produce the biotinylated p19, the optimized coding sequence was cloned into a derivative of the pHL-avitag3 vector which incorporates Avi-tag (GLNDIFEAQKIEWHE) and $His_6$ tag to the C-terminal[45], and co-transfected with the BirA biotinylating enzyme expressing plasmid (www.addgene.org) and 100 µM biotin into Expi293™ cells (Thermo Fisher Scientific)[47]. The soluble biotinylated p19 was purified from cell-free supernatant 4–5 days post-transfection using Ni Sepharose® Excel resin (Cytiva) and size exclusion chromatography (Superdex 75 Increase 10/300 GL; Cytiva) in a buffer containing 10 mM HEPES (pH 7.4), 150 mM NaCl and 3 mM EDTA. Purified biotinylated p19 was used for BLI experiments and bioassays. The extent of biotinylation was examined by SDS-PAGE gel-shift[48].

### Purification of IgGs

Recombinant IgGs were transiently expressed in Expi293F™ cells (Thermo Fisher Scientific) as per manufacturer's recommendations. The Heavy- and light-chain-coding plasmids were co-transfected at a 1:1 ratio. The antibody was purified from cell-free supernatant 4–5 days post-transfection using Protein A agarose resin (GoldBio) according to the manufacturer's recommendations and size exclusion chromatography (Superdex 200 Increase 10/300 GL; Cytiva) in a phosphate buffered saline. For GIA, purified antibodies were sterile filtered (0.22 µm), buffer exchanged into RPMI1640, and concentrated with Amicon ultra centrifugal filters (MWCO 30 kDa, Millipore Sigma). The IgG concentration was adjusted to 25 to 30 mg/ml in RPMI 1640 and aliquots were stored at −20 °C.

### ELISA

Qualitative Ab binding ELISAs were carried out as described previously[49]. Briefly, the 3D7 allele of full-length MSP-1 and p19 was coated on MaxiSorp flat-bottom 96-well ELISA plates (Nunc, Cat# 44-2404-21) at 20 µg/ml in 100 µl at 4 °C overnight. The plates were then washed thrice with phosphate buffered saline (PBS) containing 0.05% Tween 20 (PBS/T) and blocked with 2% bovine serum albumin in PBS/T for 1 h at room temperature. Next, 200 µl of 0.250 µg/ml human Ab (test, primary) in blocking buffer (PBS/T with 2% bovine serum albumin) was added to each well and incubated for 1 h at room temperature, then washed thrice with PBS/T. 200 µl of goat anti-human Ab conjugated to HRP (secondary, Jackson ImmunoResearch, Cat# 109-035-098) was then added to each well at a 1:5000 dilution and incubated for 1 h at room temperature. The plates were then washed thrice with PBS/T and developed with 70 µl of TMB substrate. The colored reaction was then stopped by adding 2 M sulfuric acid ($H_2SO_4$) and an absorbance measured at 450 nm on a BioTek™ Synergy H1 microplate reader using Gen5 3.08.01 software.

### Fab preparation and purification

Plasmids encoding Fab were synthesized by GenScript by cloning VL and VH regions into a derivative of the pHL-sec expression vector[45] upstream of the human CH, Cκ, or Cλ regions and expressed in Expi293F™ cells (Thermo Fisher Scientific) as described above. Briefly, the Fabs were purified from cell-free supernatant 4–5 days post-transfection using Ni Sepharose® Excel resin (Cytiva) and size exclusion chromatography (Superdex 200 Increase 10/300 GL; Cytiva) in a buffer containing 20 mM Tris (pH 8.0) and 100 mM NaCl. Purified Fabs were used for Biolayer Interferometry (BLI) experiments and crystallization.

### Binding affinity measurements and epitope binning using biolayer interferometry (BLI)

Binding affinity of the p19 to the Fab fragment of identified hmAbs was measured by kinetic experiments performed on an Octet RED96e (FortéBio). All measurements were performed at 200 µl/well in 10 mM HEPES (pH 7.4), 150 mM NaCl, 3 mM EDTA, 0.005% v/v surfactant P20 at 25 °C in 96-well black plates (Greiner Bio-One, Cat# 655209). Streptavidin (SA) biosensors (FortéBio, Cat# 18-5019) were used to immobilize 1.0 to 1.2 binding (nm) units of p19 (25 nM, enzymatically biotinylated on a C-terminal AviTag). Assay was performed in five sequential steps: Step 1, biosensor hydration and equilibration (630 s); Step 2, immobilization of biotinylated p19 on a Streptavidin (SA) biosensor (300 s); Step 3, wash and establish baseline (60 s); Step 4, measure p19-Fabs association kinetics (600 s); and Step 6, measure p19-Fabs dissociation kinetics (1200 s or 1500 s). The acquired raw data was processed and fit to a 1:1 binding model in order to obtain values of $K_D$, $k_a$, and $k_{dis}$ using FortéBio Data Analysis Software.

For epitope binning studies, 25 nM biotinylated p19 was captured onto Streptavidin (SA) biosensors. The kinetic assays were performed in six sequential steps: step 1, biosensor hydration and equilibration (630 s); step 2, immobilization of biotinylated p19 (300 s) ; step 3, wash and establish baseline (60 s); step 4, test Abs (Ab bin, first, 150 nM, 600 s); step 5, wash and establish baseline (60 s); and step 6, Ab binding in relation to the test Ab (second, 75 nM, 300 s). The acquired data were processed using FortéBio Data Analysis Software. The antibody pairs were analyzed for competitive binding.

### Growth inhibition assay (GIA)

GIA was performed as described in the protocol of the International Growth Inhibition Assay Reference Centre at the National Institutes of Health[50,51]. Synchronized *Plasmodium falciparum 3D7* cultures at the late schizont stage were adjusted to 1.5% parasitemia, 4% hematocrit and 20 µl aliquots were added into 96 well flat bottom tissue culture plates. A Pfs48/45 specific humanized mAb TB31F[52] was used as a negative control. Test antibody was added in triplicate wells over a concentration range from 1.0 to 0.0039 mg/ml (two-fold dilution series) and returned to culture (5% $O_2$–5% $CO_2$–90% $N_2$ at 37 °C) for 40 h. Growth inhibition (parasitemia) was assessed by the lactate dehydrogenase activity assay[53]. The percent GIA was calculated using as: % GIA = 100−100 (sample $A_{650}$ − uninfected RBC $A_{650}$)/(infected control $A_{650}$ − uninfected RBC $A_{650}$).

 

## Protein crystallization, data collection, and structure solution

For all complexes, p19 was incubated with a twofold molar excess of Fab on ice for 30 min and the complex was purified by size exclusion chromatography (Superdex 200 Increase 10/300 GL; Cytiva) in 20 mM Tris (pH 8.0) and 100 mM NaCl. Crystallization experiments were carried out using hanging drop vapor diffusion. Crystallization conditions for all complexes were obtained from crystallization trials using mosquito® crystal (SPT Labtech) by mixing 200 nl of purified complex (20.0 mg/ml) with 200 nl reservoir solution in 96-well plates at 18 °C. p19 in complex with 42D6 Fab at 20 mg/ml was crystallized with 0.2 M Potassium chloride and 20% PEG 3350 at 18 °C. p19 in complex with 42C11 Fab at 20 mg/ml was crystallized with 0.1 M HEPES (pH 7.5), 10% (w/v) PEG 4000, and 20% (w/v) Isopropanol at 18 °C. Similarly, p19 in complex with 42C3 Fab at 20 mg/ml was crystallized with 0.1 M Sodium Cacodylate (pH 6.5), 5% (v/v) PEG 8000, and 40% (v/v) (±)-2-Methyl-2,4-Pentanediol at 18 °C. All crystals were cryoprotected with the addition of either 30% glycerol or 30% polyethylene glycol. Diffraction data for all crystals were collected at beamline SER-CAT 22-ID at the Advanced Photon Source (APS). All diffraction data were processed and scaled with XDS[54] and XSCALE[54] (version February 5, 2021) and all structures were solved by molecular replacement (MR) using Phaser[55], rebuilt with AutoBuild[56], and followed by manual building in Coot[57] and refined with Phenix.refine[58]. Resolution cutoffs for scaling were evaluated using standard metrics of signal to noise and CC½. Standard settings in Phenix.refine, TLS parameters[59], B-factors, and weight optimization options (X-ray/stereochemistry weight and X-ray/ADP weight) were enabled for the refinement of the antigen-Fab complexes. The crystal structure of p19-42D6 Fab complex was solved by molecular replacement using IMC-11F8 Fab (PDB ID: 3B2U, https://www.rcsb.org/structure/3B2U) and *Pf*MSP1-19 (PDB ID: 1OB1, https://www.rcsb.org/structure/1OB1) as search models resulting in initial $R_{work}/R_{free}$ values of 0.2473/0.2934 and $R_{work}/R_{free}$ of 0.2249/0.2660 after final refinement. The crystal structure of p19-42C11 Fab complex was solved by molecular replacement using B7-15A2 (PDB ID: 1AQK, https://www.rcsb.org/structure/1AQK) and *Pf*MSP1-19 (PDB ID: 1OB1) as search models resulting in initial $R_{work}/R_{free}$ values of 0.1950/0.2284 and $R_{work}/R_{free}$ of 0.1752/0.2001 after final refinement. The crystal structure of p19-42C3 Fab complex was solved by molecular replacement using 29H4-16 (PDB ID: 6UMX, https://www.rcsb.org/structure/6UMX) and *Pf*MSP1-19 (PDB ID: 1OB1) as search models resulting in initial $R_{work}/R_{free}$ values of 0.2895/0.3265 and $R_{work}/R_{free}$ of 0.2093/0.2329 after final refinement. Crystallography data and refinement statistics are reported in Table 2. Figures of molecular structures were generated using The PyMOL Molecular Graphics System, Version 2.5 Schrödinger, LLC. Software used in this project was curated by SBGrid[60].

## Reporting summary

Further information on research design is available in the Nature Research Reporting Summary linked to this article.

## Data availability

All data generated or analysed during this study are included in this published article, source data file and supplementary information files. Atomic coordinates and structure factors have been deposited in the Protein Data Bank with PDB IDs 8DFG, 8DFH, and 8DFI. Source data are provided with this paper.

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

## Acknowledgements

This work was supported by the Intramural Research Program of the Division of Intramural Research, National Institute of Allergy and Infectious Diseases (NIAID), National Institutes of Health (NIH). The GIA activity was also supported by USAID. We thank the staff members of SER-CAT

beamline at the Advanced Photon Source (APS), Argonne National Laboratory (ANL) for beamline support. This study used the Office of Cyber Infrastructure and Computational Biology (OCICB) High Performance Computing (HPC) cluster at the National Institute of Allergy and Infectious Diseases (NIAID), Bethesda, MD. The authors would like to thank J. Patrick Gorres (LMIV, NIAID) for assistance with manuscript editing.

## Author contributions

N.H.T. and P.N.P. conceived the study. N.H.T. and P.N.P. conceived the structural, biophysical, ELISA, epitope binning, and polymorphism analyses. N.H.T., P.N.P., K.M., and C.A.L. conceived the functional analysis of hmAbs. P.N.P., T.H.D., W.K.T., K.M., and A.D. performed experiments and analyzed the data. P.D.C. and C.S.H. provided hmAb sequences. N.H.T., P.D.C., K.M., and C.A.L. supervised the studies and analyzed the data. P.N.P. and N.H.T. wrote the manuscript, with input from all authors.

## Funding

## Competing interests

N.H.T., P.N.P., C.A.L., and K.M. are listed as inventors on a provisional patent application related to this work. The remaining authors declare no competing interests.
