## [Peer Review File · Nature Communications]

Neutralizing and interfering human antibodies define the structural and mechanistic basis for antigenic diversionREVIEWER COMMENTS

Reviewer #1 (Remarks to the Author):

Patel et al., report the isolation and characterisation of neutralising and interfering antibodies against MSP-1, a malaria vaccine antigen, which has extensively researched in the malaria field but past studies on its use as vaccine were hampered by the size of the protein (~ 196 kDa) leading to discouraging results from two challenge trials in humans (Ogutu et al, 2009) and Sheey et al, 2021). Additionally, one of the problems in producing it in significant amounts recombinantly, which, however, was recently overcome in a paper published in Nature by Blank et al. 2020. Surprisingly, these articles are not referenced at all by the authors, and it is recommended to include them.

The authors focused on one of the fragments, named p19, which is the result of proteolytic cleavage of MSP-1 to the C-terminal 19 kDa fragment, which has been shown in the past to elicit neutralising antibodies. A panel of mAb was isolated and characterised biophysically and some of them as p19-mAb complexes.

Some mAbs are not neutralising, which is not surprising, whilst one of the shows broadly neutralising properties, which is indeed interesting and definitely constitute a novel insight on MSP-1 immunogenicity.

However, there are a few questions, which need to be answered by the authors before the article can be considered for publication:

- mAb were tested by GIA to evaluate growth inhibition effects. However, for completeness, in Figure 4a, 42D6 alone should be included. This is to exclude beneficial synergic effects from using 42C3 or 42C11. In addition, the 3 mAbs should also be used together in GIA experiments, as it has been done for other malaria antigens (Alanine et al., 2020), as multiple synergic effects may have been missed.

- SPR data of p19-mAbs interaction (Supplementary Figure 2) should be repeated for much longer time (i.e. 3000 sec or higher) to appreciate the difference across the antibodies with longer k_{off} .

The manuscript relies a lot on the structural data but additional information about material and methods and additional processing should be done, to be able to judge data processing and data quality. Also, a pre-deposition validation report for the mAb-p19 complexes should be included, otherwise it is impossible for reviewers to judge the quality of the presented complex structures.

The Materials and Methods section is extremely short and this is extremely surprising: "All diffraction data were processed and scaled with XDS and XSCALE45 and all structures were solved by molecular replacement (MR) using Phaser46, rebuilt with Coot47 and refined with PHENIX".

The methods should be expanded to include: resolution cut off for scaling, searching models used for molecular replacement with the initial R_{work}/R_{free} after MR for each complex and at the end of refinement. The choices of geometry constraints and B factor refinement should also be stated.

- Supplementary Figure 4 should report the density of a composite OMIT map rather than $2F_o - F_c$. Please also use r.m.s.d level instead of sigma.

Regarding the crystallographic data, I also have a few concerns:

- 1) The p19-42C11 Fab and p19-42C3 complexes have extremely tight geometry with r.m.s.d bond length of 0.003Å and resulting in unusual R_{work}/R_{free} tight convergence. Can the authors comment on why this decision was made and what happens to R_{work}/R_{free} , when geometry is relaxed during refinement? Such a tight geometry may suggest some underlying issues but it is impossible to judge without a pre-deposition report.

- 2) Data completeness in the high-resolution shell of p19-42C11 complex is rather low. Is this due

to data anisotropy? If yes, it should be commented. However, CC ½ values seems to support the author choice of resolution cut-off. Could the author please state CC 1/2 as values between 0 and 1.0 as this is more conventional?

3) Data redundancy in the crystallographic data table should be cut to the second decimal

Reviewer #2 (Remarks to the Author):

Antibodies binding to the P19 C-terminal region of Plasmodium falciparum MSP1 were epitope-mapped and their activity was assessed in a parasite growth-inhibition assay. One antibody efficiently neutralized parasite growth but others did not. These non-inhibitory antibodies effectively competed for binding to the antigen with the neutralizing antibody, abolishing its activity. The technical aspects of the study are sound, and the data provide confirmation of similar earlier work with the same antigen but different antibodies and methodology.

Minor points:

1. The use of the term "antigenic diversion" in the title does not convey a correct meaning based on the data provided.
2. BLI should be defined on first usage.
3. Lines 183/184: units of concentration should be provided.
4. Several references are incomplete.

Major point:

The authors suggest that "These findings demonstrate a generalizable pathogen immune evasion mechanism through interfering antibodies elicited by antigenic diversion..." The authors mistakenly present this as a novel concept, which it is not, and fail to cite relevant literature in the abstract and introduction.

For example, <https://pubmed.ncbi.nlm.nih.gov/11292349/> in particular, and other publications such as <https://pubmed.ncbi.nlm.nih.gov/15641776/> and <https://pubmed.ncbi.nlm.nih.gov/17511516/> that are highly relevant. <https://pubmed.ncbi.nlm.nih.gov/19627632/> is a comprehensive review from 2009 of earlier work in this area in which the concept is discussed.

The claims in the summary should be modified to indicate that the data in the current manuscript confirm the concept and the introduction should be modified to include reference to the earlier relevant work.

Reviewer #1 (Remarks to the Author):

Comment: Patel et al., report the isolation and characterisation of neutralising and interfering antibodies against MSP-1, a malaria vaccine antigen, which has extensively researched in the malaria field but past studies on its use as vaccine were hampered by the size of the protein (~ 196 kDa) leading to discouraging results from two challenge trials in humans (Ogutu et al, 2009) and Sheehy et al, 2021). Additionally, one of the problems in producing it in significant amounts recombinantly, which, however, was recently overcome in a paper published in Nature by Blank et al. 2020. Surprisingly, these articles are not referenced at all by the authors, and it is recommended to include them.

Response: We thank the reviewer for identified additional studies that should be cited, and we have revised the manuscript to include Ogutu et al., 2009; Sheehy et al., 2011; Sheehy et al., 2012; and Blank et al., 2020.

Comment: The authors focused on one of the fragments, named p19, which is the result of proteolytic cleavage of MSP-1 to the C-terminal 19 kDa fragment, which has been shown in the past to elicit neutralising antibodies. A panel of mAb was isolated and characterised biophysically and some of them as p19-mAb complexes.

Some mAbs are not neutralising, which is not surprising, whilst one of the shows broadly neutralising properties, which is indeed interesting and definitely constitute a novel insight on MSP-1 immunogenicity.

Response: We thank the reviewer for their positive comments.

Comment: However, there are a few questions, which need to be answered by the authors before the article can be considered for publication:

- mAb were tested by GIA to evaluate growth inhibition effects. However, for completeness, in Figure 4a, 42D6 alone should be included. This is to exclude beneficial synergic effects from using 42C3 or 42C11.

Response: We apologize that the 42D6 data was not immediately evident. The requested data for 42D6 alone in the GIA is included in Figure 3C preceding Figure 4. We have made a reference to this in the revised manuscript. From this data it is evident that there are no beneficial effects from using 42C3 or 42C11 in combination.

Comment: In addition, the 3 mAbs should also be used together in GIA experiments, as it has been done for other malaria antigens (Alanine et al., 2020), as multiple synergic effects may have been missed.

Response: In the example cited by the reviewer (Alanine et al., 2020), the mAbs showed positive synergistic effects and a case was made for additional combinations beyond two mAbs having additional effects. Regrettably, triple combinations of mAbs are not suitable in this case as there is no positive synergistic effect for 42C11 or 42C3 with 42D6. In addition, 42C11 and 42C3 compete for the same epitope so the triple combination once again is possible to evaluate.

Comment: - SPR data of p19-mAbs interaction (Supplementary Figure 2) should be repeated for much longer time (i.e. 3000 sec or higher) to appreciate the difference across the antibodies with longer koff.

Response: We thank the reviewer for this suggestion and agree with the reviewer that the dissociation times in the original manuscript may have been too short. Standards of practice for BLI include ensuring the dissociation time is sufficiently long to observe greater than 5% dissociation while ensuring high standards for fits to the data (Application Note: Biomolecular Binding Kinetics Assays on the Octet® Platform, 2021, Sartorius). In the revised version, we have carried out p19-mAbs binding kinetics with dissociation step long enough to observe differences in decay of binding response. The dissociation step is measured for 600 seconds for lower affinity antibodies and 900 seconds for higher affinity antibodies as more than 5% of complex dissociates within that time frame which should be sufficient for robust analysis. Kinetic measurements with longer dissociation times did not improve the fit to the data indicating the 600 and 900 second time frames were optimal for analysis. The revised data are presented in Table 1, Figure 3, Supplementary Figure 2, Supplementary Figure 3, Supplementary Table 5 and changes to the text. The revised results do not change the inferences or conclusions presented in the original or modified manuscript.

Comment: The manuscript relies a lot on the structural data but additional information about material and methods and additional processing should be done, to be able to judge data processing and data quality. Also, a pre-deposition validation report for the mAb-p19 complexes should be included, otherwise it is impossible for reviewers to judge the quality of the presented complex structures.

Response: We apologize that the pre-deposition validation report was not included. We have included this in the revised submission.

Comment: The Materials and Methods section is extremely short and this is extremely surprising:

“All diffraction data were processed and scaled with XDS and XSCALE45 and all structures were solved by molecular replacement (MR) using Phaser46, rebuilt with Coot47 and refined with PHENIX”.

The methods should be expanded to include: resolution cut off for scaling, searching models used for molecular replacement with the initial Rwork/Rfree after MR for each complex and at the end of refinement. The choices of geometry constraints and B factor refinement should also be stated.

Response: We apologize that method section for “Protein crystallization, data collection and structure solution” was short. We have expanded the methods sections to address these points.

Comment: - Supplementary Figure 4 should report the density of a composite OMIT map rather than 2Fo-Fc. Please also use r.m.s.d level instead of sigma.

Response: We thank the reviewer for identifying ways to improve the manuscript. A 2mFo-DFc composite omit map is included in the revised manuscript along with 2Fo-Fc map in supplementary Figure 4 that demonstrate the high quality of the structure determination. We have also included the r.m.s.d. levels for all maps.

Comment: Regarding the crystallographic data, I also have a few concerns:

1) The p19-42C11 Fab and p19-42C3 complexes have extremely tight geometry with r.m.s.d bond length of 0.003A and resulting in unusual Rwork/Rfree tight convergence. Can they authors comment on why this decision was made and what happens to Rwork/Rfree, when geometry is relaxed during refinement? Such a tight geometry may suggest some underlying issues but it is impossible to judge without a pre-deposition report.

Response: The geometry constraints were optimized during refinement leading to acceptable values. Indeed, lower r.m.s.d. bond lengths are more indicative of the expected bond lengths for structures. It is unclear why a low value is of concern. Relaxation of the geometry restraints as the reviewer suggests is not warranted as this leads to improvements in R work but not R free indicating overfitting of the refinement.

Again, we apologize the prevalidation report was not available during the initial submission. We have added the prevalidation report with the revised submission, and this report demonstrate no issues with the structure.

Comment: 2) Data completeness in the high-resolution shell of p19-42C11 complex is rather low. Is this due to data anisotropy? If yes, it should be commented. However, CC ½ values seems to support the author choice of resolution cut-off. Could the author please state CC 1/2 as values between 0 and 1.0 as this is more conventional?

Response: We thank the reviewer for recognizing that the CC ½ supports our choice of resolution cut-off. The data are not anisotropic, rather the slight reduction in completeness of the high-resolution shell is because the final resolution limit was higher

than expected during data collection and some of the high-resolution data lies at the edge of the detector.

Comment: 3) Data redundancy in the crystallographic data table should be cut to the second decimal

Response: We now report data redundancy to the second decimal in Table 2.

Reviewer #2 (Remarks to the Author):

Comment: Antibodies binding to the P19 C-terminal region of Plasmodium falciparum MSP1 were epitope-mapped and their activity was assessed in a parasite growth-inhibition assay. One antibody efficiently neutralized parasite growth but others did not. These non-inhibitory antibodies effectively competed for binding to the antigen with the neutralizing antibody, abolishing its activity. The technical aspects of the study are sound, and the data provide confirmation of similar earlier work with the same antigen but different antibodies and methodology.

Response: We thank the reviewer for the positive comments.

Comment: Minor points:

1. The use of the term “antigenic diversion” in the title does not convey a correct meaning based on the data provided.

Response: We thank the reviewer for identifying that the title is not correct. The title has been changed to reflect the data presented to:

“Neutralizing and interfering human antibodies define the structural and mechanistic basis for antigenic diversion”

Comment: 2. BLI should be defined on first usage.

Response: BLI has been defined on first usage in revised manuscript.

Comment: 3. Lines 183/184: units of concentration should be provided.

Response: Units of concentration are provided in the revised manuscript.

Comment: 4. Several references are incomplete.

Response: We apologize to the reviewer that references were incomplete. We have extensively revised the manuscript to rectify any incomplete references and include additional references requested by both reviewers.

Comment: Major point:

The authors suggest that “These findings demonstrate a generalizable pathogen immune evasion mechanism through interfering antibodies elicited by antigenic diversion...” The authors mistakenly present this as a novel concept, which it is not, and fail to cite relevant literature in the abstract and introduction.

For example, <https://pubmed.ncbi.nlm.nih.gov/11292349/> in particular, and other publications such as <https://pubmed.ncbi.nlm.nih.gov/15641776/> and <https://pubmed.ncbi.nlm.nih.gov/17511516/> that are highly relevant. <https://pubmed.ncbi.nlm.nih.gov/19627632/> is a comprehensive review from 2009 of earlier work in this area in which the concept is discussed.

The claims in the summary should be modified to indicate that the data in the current manuscript confirm the concept and the introduction should be modified to include reference to the earlier relevant work.

Response: We apologize to the reviewer that these important studies were not included in the initial submission. We have revised the abstract and introduction as requested to include these citations and to introduce the concept of antigenic divergence in the introduction and at the beginning of the abstract.

In addition, we have revised the last sentence of the abstract to emphasize what is novel in this manuscript, i.e. the structural and mechanistic basis for antigenic diversion from neutralizing and interfering human antibodies, as requested by the reviewer.

REVIEWERS' COMMENTS

Reviewer #1 (Remarks to the Author):

The authors have addressed all the comments about the usage of mAbs, SPR experiments and structural data in a comprehensive and satisfactory manner.
The manuscript definitely represents a step forward in the field and it is appropriate for publication in Nature communication in term of impact and audience.